# Greenhouse Soil Biosolarization with Tomato Plant Debris as a Unique Fertilizer for Tomato Crops

**DOI:** 10.3390/ijerph16020279

**Published:** 2019-01-19

**Authors:** Pablo García-Raya, César Ruiz-Olmos, José Ignacio Marín-Guirao, Carlos Asensio-Grima, Julio César Tello-Marquina, Miguel de Cara-García

**Affiliations:** 1CIAIMBITAL, Campus de Excelencia Internacional Agroalimentario, Universidad de Almería, Carretera Sacramento s/n, 04120 Almería, Spain; ceroan22@hotmail.com (C.R.-O.); jignaciomarin@gmail.com (J.I.M.-G.); casensio@ual.es (C.A.-G.); jtello@ual.es (J.C.T.-M.); 2IFAPA-La Mojonera, Camino San Nicolás n.1, 04745 La Mojonera, Spain; franciscom.cara@juntadeandalucia.es

**Keywords:** tomato, biofumigation, organic, inorganic fertilizer, sustainability, environment

## Abstract

Intensive greenhouse horticulture can cause various environmental problems. Among these, the management, storage, and processing of crop residues can provoke aquifer contamination, pest proliferation, bad odors, or the abuse of phytosanitary treatments. Biosolarization adds value to any fresh plant residue and is an efficient technique for the control of soil-borne diseases. This study aims to examine an alternative means of managing greenhouse crop residues through biosolarization and to investigate the influence of organic matter on yield and quality of tomato (*Solanum lycopersicum*, L.) fruit. With this purpose, the following nutritional systems were evaluated: inorganic fertilization with and without brassica pellets (Fert, Fert +, and Fert ++), fresh tomato plant debris with and without brassica pellets (Rest, Rest +, and Rest ++), and no fertilizer application (Control). The addition of organic matter was equal across all the treatments except for the control with regard to yield and quality of the tomato fruit. In light of these results, the application of tomato plant debris to the soil through biosolarization is postulated as an alternative for the management of crop residues, solving an environmental problem and having a favorable impact on the production and quality of tomatoes as a commercial crop.

## 1. Introduction

Protected agriculture in the Mediterranean basin has maintained a sustained growth over the last decades due to the increase of human population and the demand for vegetables. According to Castilla [1], in 2010 the total area dedicated to the cultivation of greenhouse vegetables in the Mediterranean basin reached up to 200,000 ha. The province of Almería (southeast Spain), with a protected area of 30,456 ha and commercial production of fruit and vegetables valued at 2537 M€ (tomato production corresponded to 540 M€), was considered to be the main core of protected horticultural production in Europe in 2016 [2].

However, the location of production has led to environmental problems, such as the pollution and eutrophication of aquifers, mainly due to the excessive use of pesticides, synthetic nitrogenous fertilizers, or excessive irrigation when chemical disinfection is applied [3,4,5,6,7]. Another problem is the management of crop residues due to the seasonality in the waste production [8]. In particular, in 2014 Almería produced approximately 1,900,000 tons of non-dehydrated residues from horticultural crops [8,9]. Furthermore, organic matter resources, such as green manure, mulching, animal manure, and crop waste, are frequently added to the soil through biofumigation and biosolarization to prepare the soil for the next crop [10,11].

Biosolarization [12], as a combination of biofumigation [13] and solarization [14], is a technique which can involve the application of any type of organic amendment with disinfection properties to the soil. The advantages of using the biosolarization technique include increased temperature due to the combined action of plastic sheet and the decomposition of organic matter [15,16], improved water use and soil structure [17], reduced erosion and salinity [18], increased organic matter content [19,20], organic matter solubilization [21], CO_2_ capture during the development of the biofumigant crop [22],and the acceleration of in situ decomposition of plant waste from crops which reduces the transition time between crops [23].

The increased introduction of ecological systems of production (50.9 Mha worldwide) [24], as well as the need for various organic amendments for plant nutrition, highlight greenhouse waste as viable for application through biosolarization.

The aim of this study is to evaluate the addition of organic matter (i.e., crop residues) as fertilizers and test if this organic amendment is sufficient to support profitable tomato crops grown under an intensive production system.

## 2. Materials and Methods

### 2.1. Location, Climate, and Soil

The trial was conducted in two consecutive years (2015–2016, 2016–2017) at the University of Almería-ANECOOP Experimental Research Center in Almería (36.518° N, 2.178° W). The local climate is Mediterranean arid with mild winters and hot, dry summers (average annual rainfall below 250 L·m^−2^). The experimental greenhouse was an Almería-type “raspa y amagado” greenhouse [25], the most common in the area. The greenhouse had an area of 1700 m^2^, with a northwest to southeast orientation and crops rows aligned northeast to southwest. The soil was composed of a mixture of sand and soil [26]. The history of previous crops as well as preliminary analyses showed absence of tomato soilborne pathogens in soil. During the cropping periods, no soil treatments were applied. Soil nutrition analysis was performed previously to plant transplants. Soil samples were taken at seven points throughout the greenhouse at a depth of 0–30 cm; the soil mix was analyzed by an accredited laboratory. At the start of the experiment, the soil consisted of 14.33% clay, 72.24% sand, and 13.43% silt. Soil pH was 7.56, organic matter content was 0.78%, total nitrogen (N) was 700 mg·kg^−1^, available phosphorus (P) was 61.43 mg·kg^−1^, and exchangeable potassium (K) was 365 mg·kg^−1^.

The greenhouse had a drip irrigation system with 3-L·h^−1^ emitters. In the same greenhouse during previous years (2013–2014 and 2014–2015), two tomato crops were grown with the incorporation of organic matter. In the first year, the crop was transplanted on 2 September 2015 and remained for 173 days. In the second year (2016), the crop was planted on 6 September and remained for 170 days. The plants were tomato cv. Pitenza F1 (Enza Zaden, Enkhuizen, the Netherlands) at a density of two plants per m^2^. Plants consisted of a single stem; axillary shoots were eliminated and the plant was trained along a polypropylene rope. Irrigation was performed based on readings of a Model R tensiometer (Irrometer, Riverside, CA, USA) which was placed at a depth of 30 cm; irrigation was performed at pressures between −15 and −20 KPa. Control of pests and diseases was carried out in a conventional manner according to environmental practices and legislation. Pollination was forced with the use of bumblebees at a density of four hives per ha.

Air temperature in the greenhouse was measured using a Hobo U23-001 Pro v2 temperature data logger (Onset Computer Corp., Bourne, MA, USA). During the growing period, the minimum, average, and maximum temperatures in the greenhouse were 12.60 ± 2.14 °C, 18.13 ± 1.77 °C, and 27.69 ± 3.06 °C in the first season, and 11.78 ± 4.16 °C, 18.52 ± 4.63 °C, and 30.46 ± 6.05 °C in the second season.

### 2.2. Experimental Design, Fertilization, and Soil Disinfection

The experimental design comprised seven treatments with four replications randomly distributed in two large zones (i.e., organic and inorganic). In this way, three treatments were in the inorganic zone and four in the organic zone of the greenhouse. Each elementary plot had an area of 40 m^2^, each containing 80 plants. The two zones of the greenhouse each had an independent irrigation system. In the inorganic zone of the greenhouse, the following treatments applied to the soil with inorganic fertilization were used (the nutritive solution is reported in Table 1): Nutritive solution (Fert); Nutritive solution and 0.5 kg·m^−2^ of Biofence^®^ (Fert +); Nutritive solution and 1 kg·m^−2^ of Biofence^®^ (Fert ++). In the organic area of the greenhouse, the treatments amended with different organic materials and/or exclusive irrigation with water without fertilizer were as follows: 3.5 kg·m^−2^ of fresh tomato plant debris (Rest); 3.5 kg·m^−2^ of fresh tomato plant debris and 0.5 kg·m^−2^ of Biofence^®^ (Rest +); 3.5 kg·m^−2^ of fresh tomato plant debris and 1 kg·m^−2^ of Biofence^®^ (Rest ++). The trial had a “zero” treatment (Control), which involved irrigation with water only, without the use of fertilizer or organic matter. The nutritional characteristics of the commercial product, Biofence^®^, are presented in Table 1. For the preparation of the organic amendments, fresh tomato plant debris from the previous production cycle was chopped to a particle size of less than 3 cm using tractor-powered hammer grinders and incorporated into the soil with a rototiller at the previously reported doses. The tomato debris consisted of the remaining plants at the end of the previous crop. This material included neither fruits nor roots. The existing compositional data of tomato debris is sparse and shows considerable variability among various nutrient levels [27,28,29,30]. The chemical characteristics of tomato plant debris used were: nitrogen (N) 4.12%, phosphorus (P) 0.40%, potassium (K) 2.83%, calcium (Ca) 3.43%, and magnesium (Mg) 0.86%. The commercial product, Biofence^®^, was applied along the crop row prior to the solarization of the specified treatments. All the treatments were subjected to biosolarization or solarization in the two growing periods for 60 days before transplanting the crop by covering the soil with a transparent polyethylene plastic sheet (0.05 mm thickness). The temperature was measured at 15-cm depth during the period of (bio)solarization at two points in the greenhouse using a Hobo U23-001 Pro v2 temperature probe (Onset Computer Corp., Bourne, MA, USA). Soil wetting was carried out after placing the plastic sheet using the same irrigation system, adding water up to field capacity (30 L·m^−2^). Irrigation water was analyzed during the two years of research to confirm the absence of fertilizer.

### 2.3. Parameters Analyzed

#### 2.3.1. Tomato Yield

During the growth of the crop, several parameters were measured and/or calculated for each harvest, such as yield, accumulated yield (calculated), and weight per fruit, using a Metter Toledo electronic scale. The weight per fruit was obtained from the average weight of 25 fruits with representative characteristics of the sample set. Fruits which had suitable commercial characteristics and were of the desired ripeness for consumption were harvested.

#### 2.3.2. Fruit Quality

The quality of the tomato fruit was evaluated three times in each crop cycle, using 10 marketable fruits per experimental plot (280 fruits in each of the three samplings, 840 in total). The analyzed parameters were as follows: equatorial diameter with a digital caliber (Mitutoyo); firmness of the pulp with a penetrometer (Agrosta Penefel DFT14) with an end of 0.5 cm^2^. Three measurements were taken in each fruit with gaps of 120°. Prior to measurement, the fruit cuticle was removed at each site. Fruit pH was determined with a pH-meter Crison pH-25+ with penetrating electrode. Fruit soluble solids pulp content was measured with a digital refractometer (Atago pal-1) and fruit color was measured with a colorimeter (Konica Minolta CR400). Three measurements were taken in each fruit, in three equidistant places of the equatorial zone, with gaps of 120°. The tomato color values were recorded as A*/B*.

### 2.4. Statistical Analyses

After finding that both trials could be considered statistically equal for the accumulated yield parameter, the results were analyzed as one individual experiment for a more consistent analysis. On the other hand, given that for the other tomato yield parameters (i.e., mean yield and weight per fruit) and for the quality fruit parameters the effect of year was significant and both trials could not be considered statistically equal, the results were analyzed separately. The analysis carried out for the comparisons between treatments consisted of simple analysis of variance (ANOVA) and means separated by Tukey’s honest significant difference test (*p* < 0.05). As this was a parametric analysis, the conditions of normality and homoscedasticity were checked previously (Shapiro–Wilk and Levene tests, respectively). The statistical package used was STATGRAPHIC CENTURION XVI v16.2.04 (Manugistic Incorporate, Rockville, MD, USA) for Windows.

## 3. Results

### 3.1. Tomato Yield

#### 3.1.1. Accumulated Tomato Yield

The accumulated yield during both crops (Figure 1) was consistent and did not show differences depending on the nutrition system used. Treatments with crop debris (with and without Biofence^®^) produced the same yield as plants that were fertigated (with and without Biofence^®^). All the treatments produced higher yields than the control.

#### 3.1.2. Yield per Harvest

The average yield for each harvest (Table 2) was similar to the first crop cycle (*p*-value > 0.005); the Control treatment produced substantially less throughout the cropping period, but the yield was not significantly different from the others treatments. With the second crop, differences between treatments were observed, and as occurred in the accumulated production, the treatments with crop debris (with and without Biofence^®^) produced the same yield as the fertigated treatments (with and without Biofence^®^). With the Control treatment, yield was lower with the second crop and was significantly different compared to the other treatments.

#### 3.1.3. Weight per Fruit

The weight per fruit was affected during the first growing cycle by the type of fertilization (Table 2). The treatments with inorganic fertilization produced fruits of greater weight but was not significantly different from the other nutritional treatments, except for the Rest and Control treatments. Fruits from the Rest + and Rest ++ treatment groups were of similar weight to fruits with other treatments. In the second crop, treatments with crop residues produced fruits of similar weight to those of treatments with fertigation. During the two years, control fruits had lower weights compared to any other treatment in both years of cultivation.

### 3.2. Fruit Quality

#### 3.2.1. Size

The size of the tomato fruits was smaller in the treatments with crop residues in the first crop (Table 2), although in the second crop treatments with crop residues produced fruits of similar size to those from the fertigated treatments. In both years, the Control treatment was the one that produced the smallest fruit. In all cases, the average size of the fruits was in the range of M values of 57–67 mm, which is a commercial standard.

#### 3.2.2. Firmness

The firmness of the fruits (Table 2) was affected by the treatments in both production cycles. In the first crop, the control fruits were the hardest compared to the other treatments. In the second crop, there was no difference between most of the treatments, with Fert ++, Rest, and Control being the treatments with the highest firmness values. The Fert treatment produced the softest fruits.

#### 3.2.3. Soluble Solids

The fruits from soil treated with crop residues and from the control were the sweetest in the first tomato crop. In the second crop, these differences were not apparent, with Fert ++ and Control treatments resulting in the sweetest fruits; the other treatments resulted in fruits with a similar soluble solids content (Table 2).

#### 3.2.4. Acidity of the Fruit

The fruit acidity (Table 2) was affected by the nutrition of the plants in the first crop. The control fruits had a lower pH, and the rest of the treatments resulted in fruits with very similar values, although there was a significant difference between them. In the second crop, no significant differences were observed between soil treatments (*p*-value > 0.005).

#### 3.2.5. Color

The parameter A*/B* (Table 2) showed differences between treatments in the first crop, although they were minimal and imperceptible to the human eye. The Control treatment resulted in the lowest values. In the second crop, the parameter A*/B* showed differences between treatments, which were minimal and negligible, similar to the previous year because the harvesting took place at the same point of maturity. Again, the Control treatment presented the lowest values during the second crop.

## 4. Discussion

Several authors have studied in depth the benefits for production of using techniques such as biofumigation or biosolarization in several crops [31,32,33,34,35,36,37,38,39,40,41,42].

On the other hand, there are few studies investigating plant nutrition based exclusively on the addition of organic matter applied under biosolarization in a greenhouse. Most authors supplement soil with synthetic fertilizer during the development of the crop. For this reason, it is difficult to compare past results with those from this study. The current results indicate that in both years the treatments that received organic matter did not differ from those that were fertigated, for isolated harvestings as well as for accumulated yield. Iapichino et al. [36] reported that tomatoes grown after carrying out the biosolarization technique with brassica residue (2 kg·m^−2^) and inorganic fertilization showed higher commercial production than those grown with only solarized treatments. Ros et al. [43] evaluated the biosolarization technique with various organic materials (sheep and chicken manure) and reported a greater production of pepper fruit with the use of manures; the authors did not specify whether they used an inorganic fertilizer in the culture.

Mauromicale et al. [44] reported an increase in tomato yield (up to 70% depending on the treatment) when organic matter composed of cow dung, poultry manure, and leather was incorporated into the soil prior to solarization. Again, Mauromicale et al. [16] found that the addition of compost based on cattle or horse manure prior to solarization (i.e., biosolarization) had a positive effect on the physical and chemical properties of tomato fruits in the southeast of Italy. Nuñez-Zofío et al. [45], in a trial of bio-disinfection of soils in the Basque Country, found improved production in pepper crops after the addition of various organic materials, with an increase in production of 59% with semi-composted sheep manure and poultry manure. The previous authors did not specify if they performed inorganic fertilization during the development of the crop. Marín-Guirao et al. [11] concluded that biosolarization with residues of brassicas and pellets of chicken manure supplemented with inorganic fertilization benefited tomato crops, improving the yield and organoleptic tomato fruit characteristics. The findings reported by these authors demonstrate the positive effects of biosolarization when it is supplemented with organic matter and fertigation, obtaining yields and fruits with a fruit quality comparable to a conventional system. However, the application of inorganic fertilization plus organic matter must be carried out with consideration of the global contribution of macronutrients to the system, so as to optimize resources and reduce costs. In this sense, the current research is postulated as an evolution of the techniques used by the previous authors, reducing the contribution of inorganic fertilizer to zero in the treatments with crop residues and obtaining a comparable yield to that obtained with a conventional production system.

It is necessary to analyze the control yield; this treatment was solarized in both growing years of this experiment. Solarization could be involved in the solubilization of remaining nutrients of soil, thus reaching commercial production without the addition of fertilizers. Stapleton et al. [21] found a similar effect when solarizing with transparent plastic; the authors reported an increase in the content of NO_3_^−^ and NH_4_^+^ available for post-solarization cultivation. Lombardo et al., [39] quoting Katan [14], speculated that solarization by itself can potentiate the growth and development of the plant by coining the term “increased growth response” and suggested that this is due to a rapid release of nutrients.

The use of Biofence^®^ as an organic material did not provoke an improvement either in yield or in the organoleptic properties of tomato fruit in crops over both years. Supplementation with the pellets resulted in an improvement in yield, though this change was not significant. These findings are partly consistent with those of López-Aranda et al. [46], Pane et al. [47], and Marín-Guirao et al. [11], who did not report any benefit with Biofence^®^ application. Pane et al. [47] suggested that the application of brassica carinata flour, with or without solarization, could have a protective effect on some microbial groups that benefit soil activity and the establishment of the crop. However, Guerrero et al. [48] tested Biofence^®^ for the control of nematodes of the genus *Meloidogyne* did not find any benefits from its use, as it was ineffective in controlling nematodes and had no positive effect on pepper production compared to the use of fresh manures. With regard to this study, the use of the commercial product, Biofence^®^, represented a financial investment with no corresponding increase in yield to justify its use.

The quality parameters of the tomato fruit were affected by the type of fertilization. It should be noted that the fruits from plants treated with fertigation, in the first year, had higher size and weight than fruits of the others treatments, but this was not maintained in the second year. It is noteworthy that Marín-Guirao et al. [11] obtained values for acidity, °Brix, and color similar to those obtained in this study. However, Mauromicale et al. [16] reported that parameters such as firmness, color, and soluble solids content (°Brix) were increased proportionally to the increase in organic matter in the treatments used, contrasting with the results of this study. In our study, the Control treatment showed an increase similar to that reported by Mauromicale et al. [16], which may be due to the scarcity of nutrients with no organic amendment.

From a commercial point of view, the two systems of vegetable nutrition produced fruit suitable for consumption: caliber M (57–67 mm), very high firmness (>2 kg·cm^−2^), and color between the E and F categories. It should be noted that the values of °Brix and pH were closer to a “cherry” tomato type than to a long-life tomato [49,50].

## 5. Conclusions

The incorporation of plant debris at the end of the crop cycle using biosolarization has been shown to be an efficient practice for the management of this residue, solving the problem of handling crop residues by offering a technique that respects the environment, benefits the circular economy, and provides a reference for horticultural production systems, even for the transition to organic farming. The addition of organic amendments provides the necessary nutrients for the correct development of a greenhouse tomato culture (5–6 months), achieving the same yield as a conventional inorganic fertilization system and, furthermore, maintaining the main organoleptic properties of the fruit while also being economically beneficial for growers. Future research should be focused on determining the impact on the water footprint due to improved soil structure as a result of organic matter, an aspect of vital importance for the protected agriculture of the Mediterranean basin.

## Figures and Tables

**Figure 1 ijerph-16-00279-f001:**
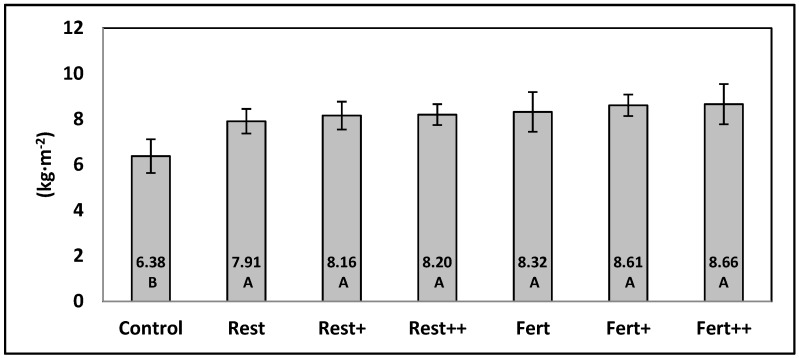
Effect of soil bio-disinfection treatments with tomato plant debris (with and without brassica pellets) as a unique fertilizer, and inorganic fertilization treatments (with and without brassica pellets) in the accumulated yield of tomato fruits. The results correspond to the average of two seasons. Different letters indicate significant differences (*p* ≤ 0.05, Tukey’s honest significant difference (HSD) test).

**Table 1 ijerph-16-00279-t001:** Nutrition systems used.

Nutritive Solution	NO_3_^−^: 11 mmol·L^−1^, H_2_PO_4_^−^: 1.5 mmol·L^−1^, SO_4_^2−^: 2 mmol·L^−1^, K^+^: 7.5 mmol·L^−1^, Ca^2+^: 5 mmol·L^−1^, Mg^2+^: 2 mmol·L^−1^.
	E.C. increased from 0.5 to 3.0 dS·m^−1^ during crop development.
Biofence^®^	Dehydrated and defatted pellets of *Brassica carinata*, 6% N, 3.1% P, 2.2% K, 1.8% S, 0.5% Mg. Triumph Italia.

**Table 2 ijerph-16-00279-t002:** Effect of soil biosolarization treatments with tomato plant debris (with and without brassica pellets) as a unique fertilizer, and of inorganic fertilization treatments (with and without brassica pellets) on tomato yield and fruit quality variables in two growing seasons (autumn–winter).

Treatment	Mean Yield (kg·m^−2^)	Fruit Weight (g)	Size (mm)	Firmness (kg·cm^−2^)	Soluble Solids (°Brix)	Fruit Acidity (pH)	Fruit Color (A*/B*)
Season 1. 2015–2016 (173 days)							
Control	0.70 ± 0.10	118.22 ± 10.06 C	61.28 ± 3.09 C	5.70 ± 0.76 A	5.31 ± 0.51 A	3.95 ± 0.12 C	0.69 ± 0.10 C
Rest	0.82 ± 0.11	123.34 ± 10.46 BC	63.02 ± 2.19 B	4.74 ± 0.68 B	5.27 ± 0.49 A	4.03 ± 0.16 B	0.73 ± 0.10 B
Rest +	0.83 ± 0.09	124.83 ± 9.97 BC	63.31 ± 2.58 B	4.45 ± 0.71 CD	5.01 ± 0.58 B	4.06 ± 0.14 AB	0.73 ± 0.10 AB
Rest ++	0.83 ± 0.12	129.97 ± 7.01 AB	63.73 ± 2.44 B	4.45 ± 0.64 CD	5.17 ± 0.42 A	4.06 ± 0.16 AB	0.73 ± 0.11 AB
Fert	0.77 ± 0.08	127.35 ± 10.22 ABC	64.68 ± 2.50 A	4.43 ± 0.65 D	4.78 ± 0.58 C	4.09 ± 0.15 A	0.73 ± 0.08 B
Fert +	0.85 ± 0.08	136.16 ± 10.28 A	65.22 ± 2.17 A	4.69 ± 0.65 BC	4.65 ± 0.44 C	4.03 ± 0.15 B	0.75 ± 0.09 A
Fert ++	0.83 ± 0.10	137.13 ± 3.93 A	65.38 ± 2.70 A	4.64 ± 0.59 BCD	4.67 ± 0.42 C	4.01 ± 0.16 B	0.74 ± 0.11 AB
*p*-value	0.6770	0.0016	0.0000	0.0000	0.0000	0.0000	0.0000
Season 2. 2016–2017 (170 days)							
Control	0.52 ± 0.07 C	106.45 ± 6.71 D	60.57 ± 3.28 D	5.65 ± 0.90 B	5.38 ± 0.55 B	4.12 ± 0.14	0.45 ± 0.11 D
Rest	0.70 ± 0.15 B	120.77 ± 12.25 C	63.97 ± 3.32 BC	5.70 ± 1.00 B	5.34 ± 0.59 B	4.13 ± 0.23	0.51 ± 0.11 C
Rest +	0.73 ± 0.13 AB	122.85 ± 11.24 BC	64.56 ± 3.26 B	5.56 ± 0.91 B	5.21 ± 0.60 BC	4.13 ± 0.11	0.51 ± 0.10 C
Rest ++	0.74 ± 0.12 AB	126.32 ± 11.50 ABC	64.07 ± 3.06 BC	5.63 ± 0.96 B	5.27 ± 0.55 BC	4.11 ± 0.10	0.52 ± 0.10 BC
Fert	0.81 ± 0.12 A	132.03 ± 5.68 A	65.64 ± 2.45 A	5.18 ± 0.89 C	5.10 ± 0.48 C	4.21 ± 0.27	0.53 ± 0.11 AB
Fert +	0.79 ± 0.12 AB	127.01 ± 6.53 AB	64.81 ± 2.84 AB	5.59 ± 0.98 B	5.28 ± 0.49 B	4.16 ± 0.12	0.52 ± 0.11 BC
Fert ++	0.82 ± 0.13 A	124.63 ± 7.42 BC	63.60 ± 2.82 C	6.05 ± 0.96 A	5.63 ± 0.51 A	4.16 ± 0.23	0.55 ± 0.11 A
*p*-value	0.0000	0.0000	0.0000	0.0000	0.0000	0.0555	0.0000

* The same letter within columns indicates no significant difference (*p* ≤ 0.05, Tukey’s HSD test).

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
