# Peer review of "Greenhouse Soil Biosolarization with Tomato Plant Debris as a Unique Fertilizer for Tomato Crops"

_ijerph, 2019, doi:10.3390/ijerph16020279_

Round 1

Reviewer 1 Report

Reuse of plant waste in agriculture and horticulture is an age old technique.  Authors re-confirming it is good.  Lines 267-269 can be eliminated - writers suggesting they should carryout proper discussions does not add substance to the paper.  

Conclusions - the way it is shown is mediocre - the authors can note the fact that 'forests' enhance and grow by reusing their own debris -  or recycling bioelements needed for regrowth.  

Author Response

Reuse of plant waste in agriculture and horticulture is an age old technique.  Authors re-confirming it is good.  Lines 267-269 can be eliminated - writers suggesting they should carryout proper discussions does not add substance to the paper.  

Conclusions - the way it is shown is mediocre - the authors can note the fact that 'forests' enhance and grow by reusing their own debris -  or recycling bioelements needed for regrowth.  

We totally agree with you that in natural ecosystems (not intervened by humans) the balance is achieved by the reutilization of resources.

As good as you specified: “'forests' enhance and grow by reusing their own debris -  or recycling bio-elements needed for re-growth”, but in an agro industrial environment this conclusion is not as obvious, the actual trend is to use as much inputs as possible to ensure the profit.

This research pretends to establish an eco-friendly approach to the greenhouse debris problematic. Further investigation must be required from an economical point of view to support the presented conclusions.

Reviewer 2 Report

The paper should be edited taking into account the suggestions in the file

Author Response

Comments and Suggestions for Authors

The paper should be edited taking into account the suggestions in the file.

All the suggestions were accepted and changed in the new version of the documents (Tracked and Clean).
